# Cobalt Phthalocyanine-Ionic Liquid Composite Modified Electrodes for the Voltammetric Detection of DNA Hybridization Related to Hepatitis B Virus

**DOI:** 10.3390/mi12070753

**Published:** 2021-06-26

**Authors:** Ece Yaralı, Arzum Erdem

**Affiliations:** 1Department of Analytical Chemistry, Faculty of Pharmacy, Ege University, Bornova, Izmir 35100, Turkey; eceyarali@hotmail.com; 2Department of Materials Science and Engineering, Graduate School of Natural and Applied Science, Ege University, Bornova, Izmir 35100, Turkey

**Keywords:** cobalt phthalocyanine, ionic liquid, voltammetric biosensor, Hepatitis B virus, pencil graphite electrode, differential pulse voltammetry

## Abstract

In this study, cobalt phthalocyanine (CoPc) and ionic liquid (IL) modified pencil graphite electrodes (PGEs) were designed and implemented to detect sequence-selective DNA hybridization related to the Hepatitis B virus (HBV). The surface characterization of CoPc-IL-PGEs was investigated by scanning electron microscopy (SEM), and the electrochemical behavior of electrodes were studied by electrochemical impedance spectroscopy (EIS) and cyclic voltammetry (CV) techniques. The voltammetric detection of hybridization was investigated by evaluating the guanine oxidation signal, measured by differential pulse voltammetry (DPV) technique. The implementation of our biosensor to serum samples was also examined using fetal bovine serum (FBS). The detection limit was established as 0.19 µg/mL in phosphate buffer solution (PBS) (pH 7.40) and 2.48 µg/mL in FBS medium. The selectivity of our assay regarding HBV DNA hybridization in FBS medium was tested in the presence of other DNA sequences. With this aim, the hybridization of DNA probe with non-complementary (NC) or mismatched DNA sequence (MM), or in the presence of mixture samples containing DNA target NC (1:1) or DNA target MM (1:1), was studied based on the changes in guanine signal.

## 1. Introduction

Biosensors are diagnostic tools that provide specific and quantitative determination of a target, e.g., nucleic acid sequences representing a genetic or epidemic disease, an anticancer drug, or an environmental pollutant [1]. Biosensors are classified according to the biological specificity conferring mechanism or the mode of physicochemical signal transduction. In addition, they may be classified according to the analytes or reactions that they monitor, for instance, the direct monitoring of analyte concentration or of reactions producing or consuming such analytes. In terms of transduction principles, biosensors can be classified as optical, electrochemical, mass, magnetic, calorimetric, micromechanical, or piezoelectrical [1,2,3].

Metallophthalocyanines (MPcs) have effective properties such as designed flexibility, various coordination properties, different substitutional alternatives high thermal and chemical stability, and attractive electrochemical properties [4,5,6,7,8,9,10,11,12,13].

Cobalt phthalocyanine (CoPc), a member of the MPcs family, has a central atom as Co^2+^ and an 18 p-electron system [14]. CoPcs are two dimensional (2D) organic macrocyclic molecular catalysts with cobalt atoms at the center. CoPc is similar to porphyrin in its structure and demonstrates certain physicochemical, unique electronic, and optical properties [15,16].

Ionic liquids (ILs) are either organic salts or mixtures of salts. ILs have excellent physicochemical properties, including wide electrochemical windows, low vapor pressure, high ionic conductivity, good antifouling ability and biocompatibility, natural catalytic capability, and thermal and chemical stability [17,18]. ILs have been extensively used as modifiers at electrode surfaces for the production of biosensors [19,20] and gas sensors [21,22] because of their excellent electrochemical properties [23].

The hepatitis B virus (HBV) induces hepatocelular carcinoma, chirrosis, and acute and chronic hepatitis [24]. The ninth highest cause of death in the world is HBV infection [25,26,27]. More than 2000 million people are infected with HBV, according to the data of the World Health Organization (WHO) [28].

The purpose of antiviral treatments of chronic hepatitis B is the continued suppression or loss of detectable HBV DNA in serum, which is monitored by nucleic acid hybridization and amplification methods [29]. An increasing demand for the determination of trace amounts of DNA exists for rapid, ultrasensitive, and reliable assays, including biosensors [30,31,32,33].

There are earlier electrochemical studies for HBV detection in the literatures [34,35]. For instance, Chen et al. [34] developed gold nanoparticles (GNPs) deposited aluminum oxide (AAO) film modified electrode for impedimetric detection of HBV DNA. The hybridization between the probe and its complementary sequence (i.e., HBV DNA target with 3020–3320-mer) in clinical samples was performed and, accordingly, the impedimetric detection was implemented for the detection of HBV DNA [36].

Mashhadizadeh et al. [35] reported DNA biosensor for the impedimetric detection of HBV DNA that was developed by using gold nanoparticles (GNP) and magnetic nanoparticles.

To our best knowledge, this is the first time that CoPc-IL-PGE has been developed as a nucleic acid biosensor and its implementation to detect HBV DNA hybridization voltammetrically under optimized conditions presented in the literature. The electrochemical behavior and the surface characterization of the electrode was investigated by using CV, EIS, and SEM techniques, respectively. The selectivity of our assay on nucleic acid hybridization was examined in the presence of NC or MM, or and in the presence of mixture samples (1:1) of DNA target NC, or DNA target MM, prepared in FBS medium.

## 2. Materials and Methods

### 2.1. Apparatus

The three-electrode system was composed of an Ag/AgCl/3 M KCl reference electrode (BAS, Model RE-5B, W. Lafayette, IN, USA), platinum wire as the auxiliary electrode, and a pencil graphite electrode (PGE) as the working electrode.

CV measurements were carried out using a µ-AUTOLAB electrochemical analysis system with GPES 4.9 software package (Eco Chemie, Utrecht, The Netherlands). DPV and EIS measurements were carried out using an AUTOLAB-PGSTAT-30 electrochemical analysis system supplied with the NOVA 1.11 software package. All measurements were carried out in a Faraday cage (Eco Chemie, Utrecht, The Netherlands).

### 2.2. Chemicals

Cobalt (II) phthalocyanine (CoPc), the ionic liquid (IL) 1-butyl-3-methylimidazolium hexafluorophosphate and fetal bovine serum (FBS) were purchased from Sigma–Aldrich. All chemicals were of analytical reagent grade and they were purchased from Sigma and Merck. Ultrapure and deionized water was used throughout.

The information about fish sperm double-stranded DNA (fsDNA, Sigma, Berlin, Germany) and DNA oligonucleotides with their stock solutions are given in detail in Appendix A.

### 2.3. Procedure

Each of the following steps was performed in order to perform the experimental procedure properly:Surface modification of PGEs with CoPc-ILs;Hybridization of HBV DNA probe with its target DNA or other oligonucleotides; NC or MM;Immobilization of DNA-DNA hybrids onto the surface of CoPc-IL-PGEs.

Information regarding the experimental procedure is also given in more details in the Appendix A.

### 2.4. Microscopic Characterization of Electrodes

The SEM technique was used for microscopic characterization of the surface of each electrode. Details on SEM characterization are presented in Appendix A.

### 2.5. Electrochemical Measurements

Electrochemical measurements were performed in order to investigate the electrochemical behavior of electrodes with their optimization studies and finally to detect HBV DNA hybridization. More details about electrochemical measurements are given in Appendix A.

## 3. Results and Discussion

Firstly, the surfaces of each of electrodes; PGE, CoPc-PGE, IL-PGE, and CoPc-IL-PGE were characterized by SEM (shown in Figure 1). The surface roughness of PGE can be seen clearly in Figure 1a,b. In the presence of CoPc modification on the electrode surface, sub-monolayers of CoPc are not observed, contrary to the microcrystalline structures. The homogenous surface after IL modification can be seen in Figure 1e,f. The most homogenous surface was observed after CoPc-IL modification onto the surface of PGE.

The concentration of CoPc was optimized based on the changes at the signal, measured by the CV technique (Figure 2). The highest anodic peak current (I_a_) was measured by using 50 µg/mL CoPc on the modification of the PGE surface. Similarly, the highest increase in I_a_, in comparison to the one from PGE, was obtained as 38.40% in the presence of 50 µg/mL CoPc with IL modified PGE. In accordance with these results, it was concluded that the most conductive surface was obtained by using 50 µg/mL CoPc to develop CoPc-IL-PGEs. Thus, 50 µg/mL was selected as the optimal concentration of CoPc for further experiments. The results of the optimization studies are listed in Table 1.

In order to investigate the electrochemical behavior of these electrodes, a batch of measurements was performed by CV and, accordingly, the voltammograms of PGE, IL-PGE, CoPc-PGE, and CoPc-IL-PGE are presented in Figure 3. The effective surface areas of the each of the electrodes were calculated according to the Randles–Sevcik Equation (Equation (1)) [37];
(1)$$I_{p}=\left(2.69\times 10^{5}\right)\cdot n^{3/2}\cdot A\cdot C\cdot D^{1/2}\cdot \upsilon^{1/2}$$
where *I_p_* = peak current; *n* = electron stoichiometry; *A* = electrode area, cm^2^; *C* = concentration, mol/cm^3^; *D* = diffusion coefficient, cm^2^/s; and *υ* = scan rate volt/sec. The surface areas of PGE, IL-PGE, CoPc-PGE, and CoPc-IL-PGEs were found to be 0.228 cm^2^, 0.291 cm^2^, 0.163 cm^2^, and 0.316 cm^2^, respectively. The increase % at the surface area of CoPc-IL-PGE in comparison with PGE was calculated and recorded as 39%. As a result, the highest surface area was obtained by CoPc-IL-PGEs.

Furthermore, the electrochemical characterization of the each of the electrodes was examined using EIS (Figure 4) during the investigating into the change of R_ct_ values before and after immobilization of a 0.5 µg/mL amino linked HBV DNA probe onto the surface of each electrode. The average R_ct_ value was measured as 66.81 ± 11.51 Ohm (RSD%, 17.22%, n = 5) using PGE. After modification of CoPc-IL onto the electrode surface, there was a decrease of R_ct_ due to the conductivity of IL that increased the charge-transfer between the solution interface and the electrode. After immobilization of the probe onto the surface of CoPc-IL-PGE, the average R_ct_ value was recorded as 680.33 ± 17.79 Ohm (RSD%, 2.61%, n = 5), which is larger than the one obtained before probe immobilization (i.e., 34.98 ± 7.99 Ohm). An increase in R_ct_ value was recorded due to the negative charges coming from the phosphate backbone of the DNA, preventing [Fe(CN)_6_]^3−/4−^ from reaching the surface of the electrode [38,39,40,41]. An increase ratio at CoPc-IL-PGE of about 19.45% was calculated in the absence/presence of probe immobilization, whereas an increase ratio at PGE, IL-PGE, and CoPc-PGE was calculated as 11.62%, 17.20%, and 4.29%, respectively.

The apparent fractional coverage value (θ_IS_^R^) described by Janek et al. [42] was calculated. Accordingly, the values of θ_IS_^R^ were found to be 0.915, 0.935, 0.744, and 0.949, respectively.

After immobilization of 2.5 μg/mL fsDNA onto the surface of PGE, IL-PGE, CoPc-PGE, and CoPc-IL-PGE, the guanine oxidation signal was measured by DPV technique, and the average signals were found to be, respectively, 2.29 ± 0.18 µA (RSD%, 8.20%, n = 3), 2.39 ± 0.35 µA (RSD%, 14.55%, n = 3), 2.21 ± 0.23 µA (RSD%, 10.45%, n = 3), and 2.53 ± 0.28 µA (RSD%, 11.12%, n = 3) (Appendix A). In accordance with these results, the highest guanine oxidation signal was obtained using CoPc-IL-PGE.

The effect of labelling DNA oligonucleotides with different chemical groups (-NH_2_, -PO_4_, or unmodified) was also investigated regarding the electrode’s response (Figure 5). After the immobilization of probe linked with the -NH_2_ group onto the surface of CoPc-IL-PGE, the highest guanine oxidation signal was obtained (16.29 ± 0.33 µA with RSD%, 2.02%, n = 3) (Figure 5B). According to these results, the highest guanine oxidation signal was obtained by using -NH_2_ linked DNA probe immobilized CoPc-IL-PGE. This was due to the fact that efficient binding of the amino linked DNA probe to the surface of CoPc-IL-PGE via the electrostatic interaction (π−π) occurs because of the interaction of the negatively charged phosphate groups of IL with the positively charged amino group of the DNA probe [43]. Therefore, a DNA probe linked with the -NH_2_ group was selected for performing the hybridization experiments related to HBV DNA detection.

The effect of the hybridization process upon the response was investigated, and the results are presented in Appendix A. The hybridization process was performed in our study by following two different procedures: (i) step by step and (ii) solution phase, in order to present the hybridization efficiency of the response, i.e., guanine oxidation signal. No guanine base is available in the probe sequence that is specific to HBV. As a consequence, this detail provides an advantage to our assay because our assay provides direct voltammetric detection of nucleic acid hybridization by measuring the oxidation signal of guanine observed at the peak potential of +1.00 V [32,44]. Accordingly, the guanine oxidation signals were measured as 8.09 ± 1.15 µA (RSD% = 14.20%, n = 3) and 8.44 ± 0.37 µA (RSD%, 4.43%, n = 3), respectively (Appendix A). Subsequently the most reproducible and the highest guanine oxidation signal was recorded by following the solution phase hybridization process; it was then chosen in our further hybridization experiments.

Next, the effect of probe concentration onto the response was studied (Appendix A). Hybridization between 5 µg/mL HBV target and DNA probe at its different concentrations, varying from 0.25 to 1 µg/mL, was executed over 30 min. After the hybrids formed between each of the DNA probe concentrations of 0.25, 0.5, 0.75 and 1 µg/mL on the HBV probe with 5 µg/mL HBV target, it was immobilized onto the surface CoPc-IL-PGEs. Then, the guanine signal was measured, as shown in Appendix A. According to these results, there was an increase of the guanine oxidation signal until 0.5 µg/mL of the probe, and then it levelled off at a concentration of the DNA probe at 0.5 µg/mL. Therefore, 0.5 µg/mL of HBV DNA probe concentration level was chosen for further hybridization experiments.

The effect of hybridization time was then investigated based on the changes in guanine signal after hybridization performed at various hybridization times, such as 3 min, 5 min, and 15 min (Appendix A). The highest guanine signal was measured (6.40 ± 0.89 µA with RSD%, 13.86%, n = 3) for the 5 min hybridization time (Appendix A). In the presence of longer hybridization times, there is a decrease observed in the hybridization response. Hence, 5 min was chosen as the optimal hybridization time in our study.

The effect of changes in HBV target concentration on the response was studied with different concentrations of target from 2.5 µg/mL to 15 µg/mL. An increase in the guanine oxidation signal was recorded from 2.5 to 12.5 µg/mL of the target, and the signal levelled off at 12.5 µg/mL. The resulting line graph is presented in Appendix A. The highest guanine signal with better reproducibility was obtained in the presence of hybridization of 0.5 μg/mL probe with 12.5 μg/mL of HBV target. The detection limit (DL) [45] was calculated and found to be 0.19 µg/mL (equal to 0.03 µM, 1.22 pmol in the 40 µL sample) in the linear concentration range from 0.5 to 2.5 μg/mL (Figure 6) using the equation I = 1.349 × C_HBV target_ –0.246 with the coefficient of determination (R^2^) of 0.99 with a sensitivity of 4.268 µA·mL/µg·cm^2^.

Additionally, voltammetric detection of HBV DNA was examined using CoPc-IL-PGE in the presence of artificial serum, fetal bovine serum (FBS). With this goal in mind, the effect of FBS dilution upon the response was first studied in different dilution ratios (1:100 or 1:500) for FBS:PBS. For both mediums, the signal was measured at the peak potential of +0.739 (Appendix A). In order to minimize the interference effect of FBS upon the guanine signal, FBS medium with a dilution ratio of 1:500 was chosen in our study for further experiments because the lowest signal was recorded at +0.739 V using this medium.

The effect of the changes at HBV target concentration upon the response was then studied in the diluted FBS medium, and the resulting line graph in target concentration varying from 5 µg/mL to 35 µg/mL is shown in Appendix A. An increase in the guanine signal was recorded from 5 to 30 µg/mL for the HBV target and then levelled off at 30 µg/mL. The detection limit (DL) [45] was calculated and found to be 2.48 µg/mL (equal to 0.4 µM, 16 pmol in the 40 µL sample) in the linear concentration range from 5 to 30 μg/mL (Figure 7) using the equation I = 0.308 × C_HBV target_ –0.088 with the coefficient of determination (R^2^) of 0.99 with the sensitivity of 0.975 µA·mL/µg·cm^2^.

Furthermore, the selectivity of our assay on hybridization in FBS medium was tested in the presence of unwanted substituents, e.g., NC, MM sequences, as well as in the mixture samples containing HBV target:NC (1:1) or HBV target:MM (1:1) (shown in Appendix A). After the hybridization between 0.5 μg/mL probe with 30 μg/mL HBV target, NC and MM, the average guanine signals were measured as 9.25 ± 0.65 µA (RSD%, 7.07%, n = 3), 3.43 ± 0.40 µA (RSD%, 11.83%, n = 3), and 6.27 ± 0.50 µA (RSD%, 8.02%, n = 3), respectively. Moreover, in the presence of the mixture of HBV target:NC (1:1) and HBV target:MM (1:1), the average guanine signals were measured as 8.13 ± 1.11 µA (RSD%, 13.64%, n = 3) and 9.33 ± 1.53 µA (RSD%, 16.40%, n = 3), respectively. In view of that, our assay based on CoPc-IL-PGE showed a very selective behavior to its target HBV DNA sequence, even in FBS matrix containing unwanted substituents.

## 4. Conclusions

To the best of our knowledge, this is the first time that CoPc-IL modified electrodes have been developed and applied to the electrochemical detection of DNA related to HBV, and presented in the literature. The SEM technique was used for the microscopic characterization of electrodes. CV and EIS techniques were used for electrochemical characterization of electrodes. Optimization studies on the hybridization process, e.g., hybridization time, the concentrations of amino linked HBV probe, and HBV target were studied by using DPV, dependent upon the changes of the guanine signal. The detection limit was calculated in both mediums in PBS (i.e., 0.19 µg/mL) and in FBS:PBS diluted solution (i.e., 2.48 µg/mL) with a sensitivity of 4.268 and 0.975 µA·mL/µg·cm^2^, respectively.

The selectivity of our assay was tested against NC, MM sequences, and against mixture samples containing complementary HBV DNA target:NC (1:1) or HBV DNA target:MM (1:1) in FBS:PBS diluted solution under optimum conditions. In accordance with these results, it was deduced that our assay showed a very selective behavior in the detection of HBV DNA hybridization, even in the presence of other oligonucleotides in contrast to target HBV DNA.

The significant properties of PGEs have provided some significant advantages, such as being robust, easy to use, being single-use, and providing a faster and more sensitive detection protocol in comparison with glassy carbon electrodes [46], gold electrodes [47,48], and carbon paste electrodes [35], which have been previously used for the detection of HBV. In contrast to the studies of Chen et al. [34] and Mashhadizadeh et al. [35], the preparation and modification of the electrode surface can be done in a shorter time (i.e., less than one hour).

Furthermore, these electrodes are cheap and easy to develop as single-use, and are portable in chip technology.

Due to the fact that inefficient hybridization prevents the full match of target nucleic acids with their capture probes [49,50], it is very critical to perform the high hybridization process. Herein, the electrochemical detection of HBV DNA by following a solution phase hybridization process yields the highest response of good reproducibility by CoPc-IL-PGEs.

In the present study, the DL was calculated as 0.19 µg/mL in buffer medium (1.224 pmol in 40 µL sample), which was found to be lower in contrast to the ones reported in previous studies [33,43,48,51]. In addition, the DL in FBS medium was also found to be 2.48 µg/mL (16 pmol in 40 µL sample). Following the procedure of solution phase hybridization in our study provides a fast, efficient, and more sensitive hybridization, resulting in a detection limit (0.19 µg/mL) that is 57 times lower in comparison to the one (i.e., 10.88 µg/mL) in the study that followed the step-by-step hybridization procedure for label free HBV DNA detection performed by CHIT-CNF-PGEs [33].

In another study based on solution phase hybridization applied to label-free HBV DNA detection by HaNPs-PGEs [51], a 5.5 times higher DL value (i.e., 1.07 µg/mL) was reported in comparison to the one (i.e., 0.19 µg/mL) achieved in the present study.As a conclusion, CoPc-IL-PGEs were designed and introduced for the first time in our study, and these electrodes were successfully implemented in the sensitive and selective detection of sequence selective DNA hybridization related to HBV. Due to the advanced properties of Cobalt phthalocyanine-ionic liquid composites accelerating the electron transfer and increasing the surface area of the electrode, in our study, the voltammetric detection of HBV DNA resulted in a lower detection limit with good selectivity, even in the FBS medium.

## Figures and Tables

**Figure 1 micromachines-12-00753-f001:**
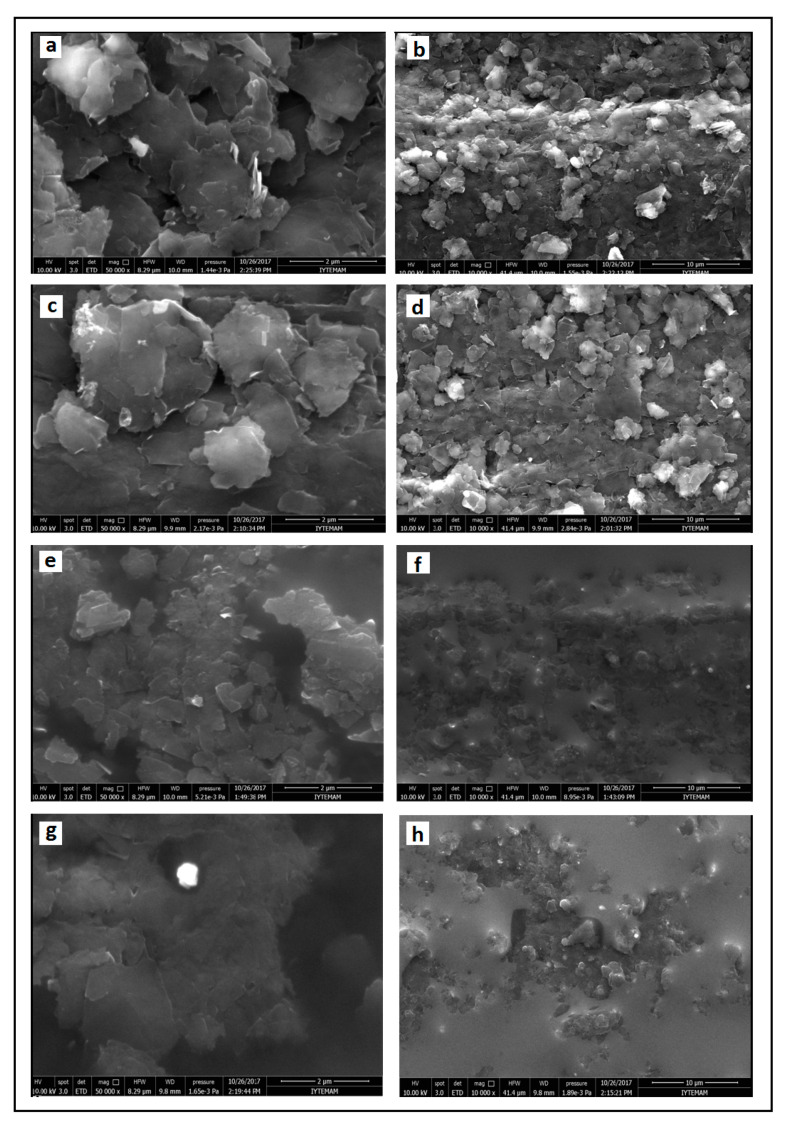
SEM images of PGE (**a**,**b**), CoPc-PGE (**c**,**d**), IL-PGE (**e**,**f**), and CoPc-IL-PGE (**g**,**h**) using the identical acceleration voltage of 10 kV with a resolution of 2 µm and 10 µm, respectively.

**Figure 2 micromachines-12-00753-f002:**
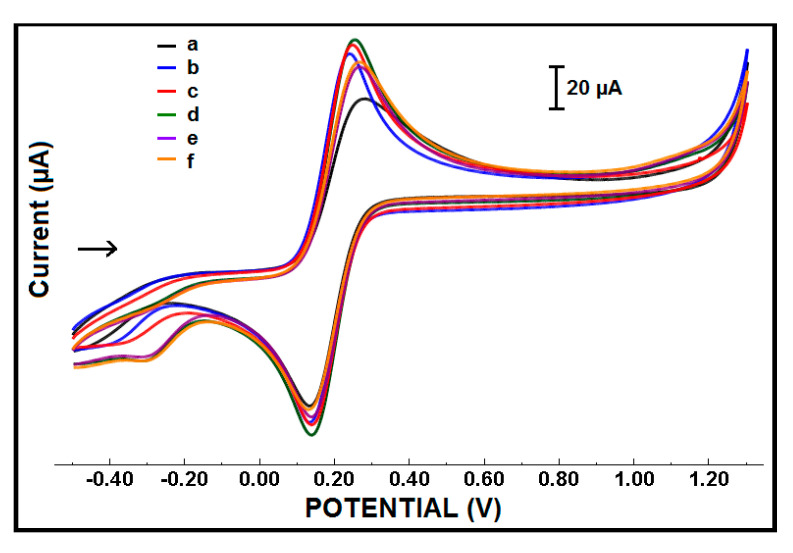
Cylic voltammograms of (a) PGE, (b) ILPGE, (c) 25 µg/mL CoPc-IL-PGE, (d) 50 µg/mL CoPc-IL-PGE, (e) 100 µg/mL CoPc-IL-PGE, and (f) 200 µg/mL CoPc-IL-PGE in the presence of 5% IL.

**Figure 3 micromachines-12-00753-f003:**
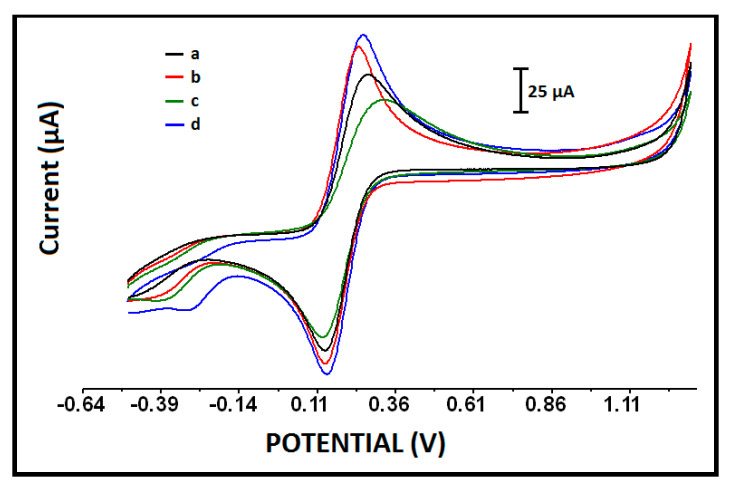
Cylic voltammograms of (a) PGE, (b) IL-PGE, (c) CoPc-PGE, and (d) CoPc-IL-PGE under the optimized conditions: 50 µg/mL CoPc and 5% IL modified PGE.

**Figure 4 micromachines-12-00753-f004:**
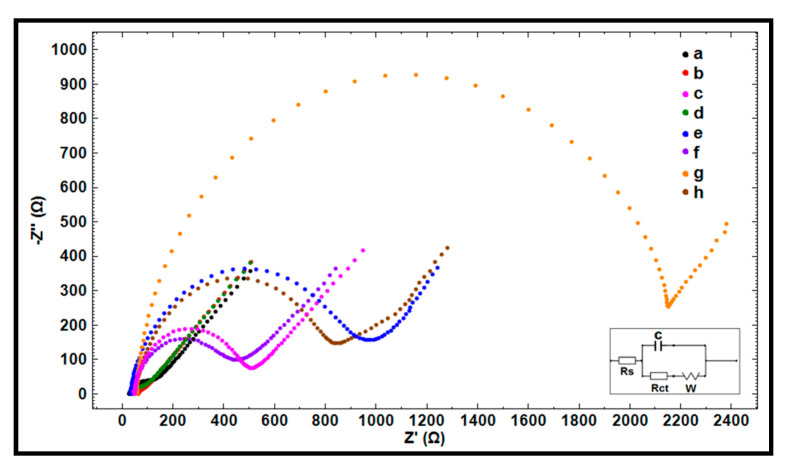
Nyquist diagrams representing the R_ct_ values obtained by (a) PGE, (b) IL-PGE, (c) CoPc-PGE, and (d) CoPc-IL-PGE; 0.5 µg/mL amino linked HBV DNA probe immobilized (e) PGE, (f) IL-PGE, (g) CoPc-PGE, and (h) CoPc-IL-PGE. Inset was the equivalent circuit model used for fitting the impedance data, the parameters of which are listed in the text; R_s_ is the solution resistance. The constant phase element, Q, is then related to the space charge capacitance at the electrode/electrolyte interface. R_ct_ is related to the charge transfer resistance at the electrode/electrolyte interface. The constant phase element, W, is the Warburg impedance due to mass transfer to the electrode surface.

**Figure 5 micromachines-12-00753-f005:**
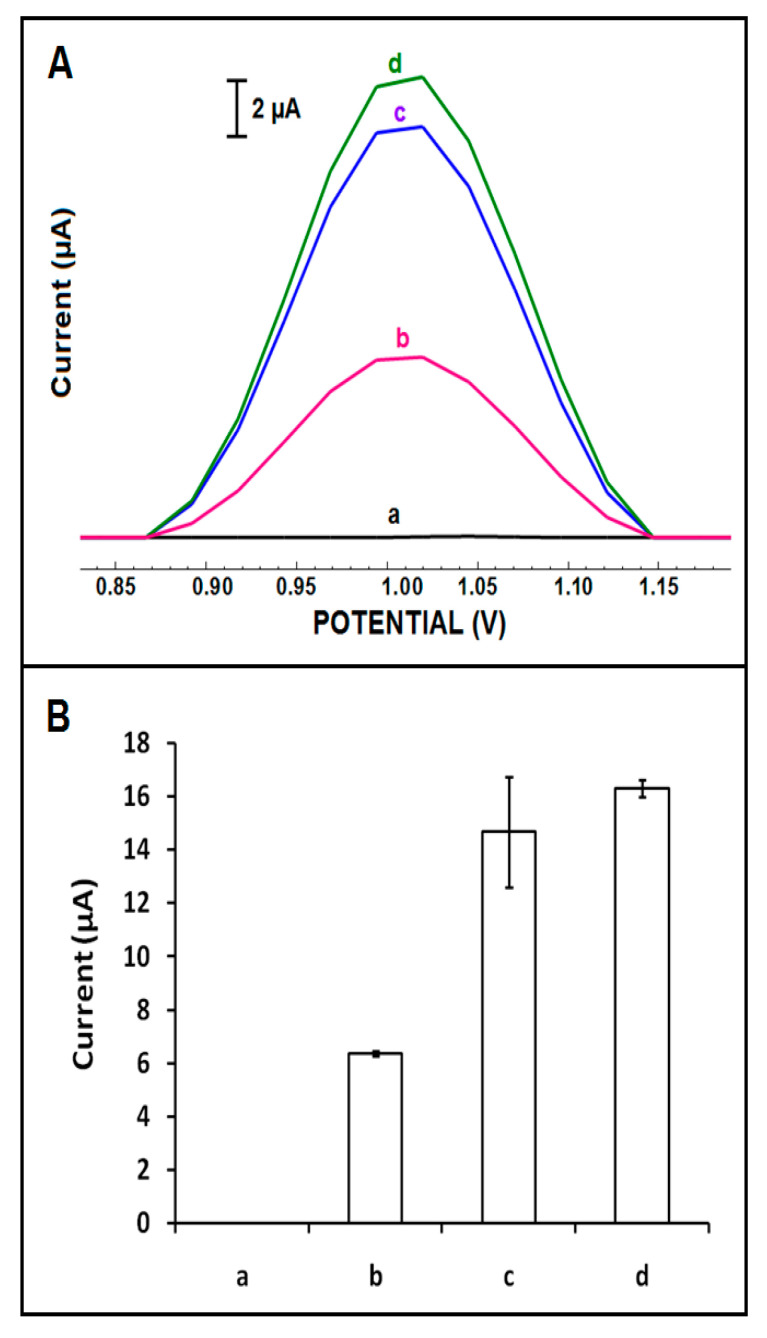
(**A**) DPVs and (**B**) histograms representing control experiment of (a) CoPc-IL-PGE in ABS, the guanine oxidation signals of 5 µg/mL, (b) bare (unmodified) DNA probe, (c) PO_4_ linked DNA probe, and (d) NH_2_ linked DNA probe immobilized CoPc-IL-PGE in the presence of hybridization with its complementary target DNA.

**Figure 6 micromachines-12-00753-f006:**
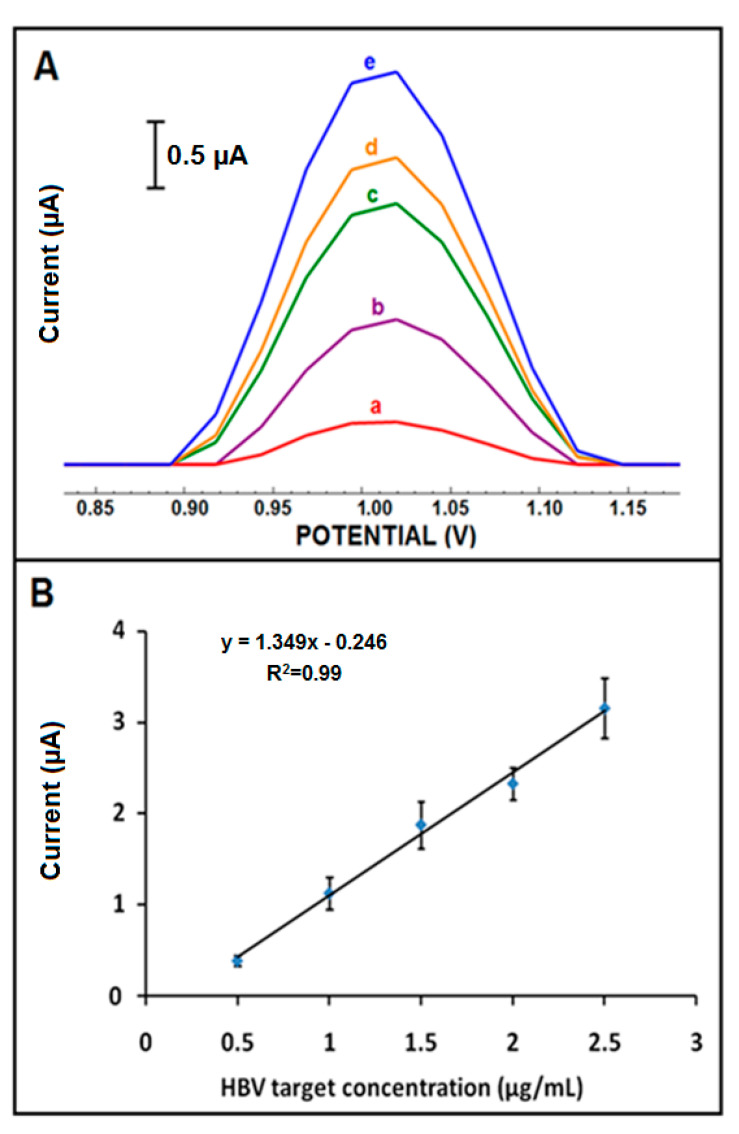
(**A**) DPVs represent the average guanine oxidation signals (n = 3) after hybridization between amino linked HBV probe and (a) 0.5, (b) 1, (c) 1.5, (d) 2, and (e) 2.5 µg/mL HBV target onto the surface of CoPc-IL-PGE (n = 3). (**B**) Calibration plot obtained from oxidation signals of guanine after hybridization of amino linked HBV probe with different concentrations of HBV target at the surface of CoPc-IL-PGE.

**Figure 7 micromachines-12-00753-f007:**
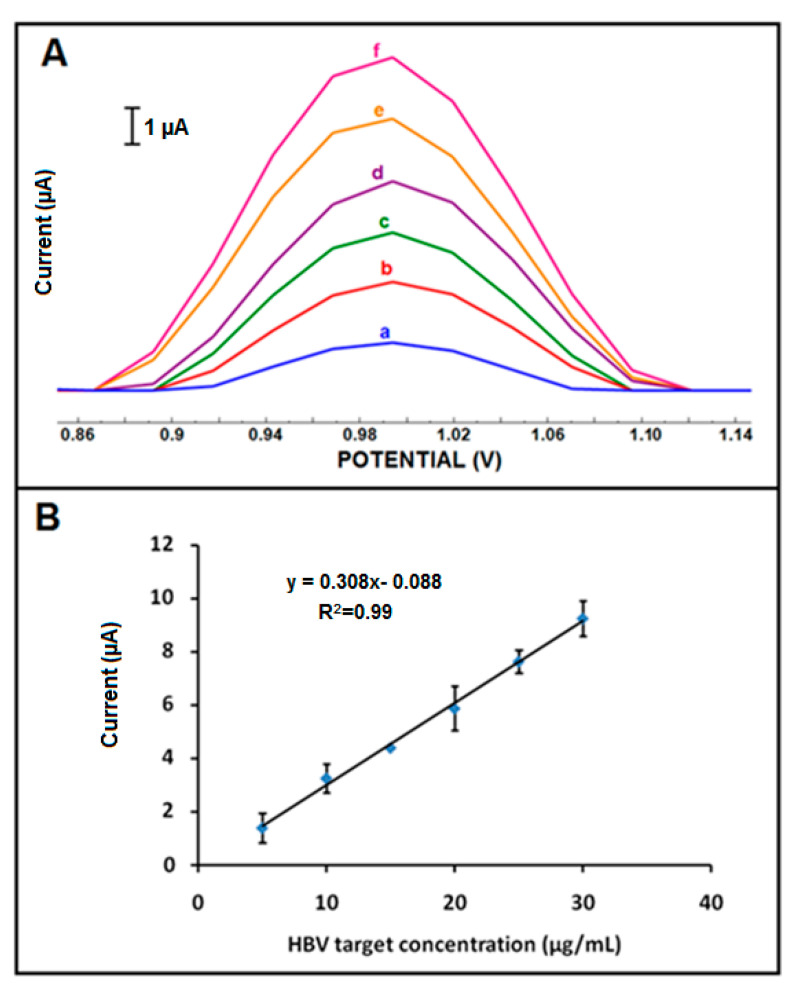
(**A**) DPVs represent the average guanine oxidation signals (n = 3) after hybridization between amino linked HBV probe and (a) 5, (b) 10, (c) 15, (d) 20, (e) 25, and (f) 30 µg/mL HBV targets in FBS medium onto the surface of CoPc-IL-PGE. (**B**) Calibration plot obtained from guanine oxidation signals after hybridization of HBV probe with different concentrations of HBV target in FBS medium onto the surface of CoPc-IL-PGE.

**Table 1 micromachines-12-00753-t001:** The average values (n = 3) of anodic peak current (I_a_), the cathodic current (I_c_), the anodic charge (Q_a_), and the cathodic charge (Q_c_) obtained by using PGEs, IL-PGEs, and CoPc-IL modified PGEs in various concentration of CoPc.

Electrode	I_a_ (µA)	I_c_ (µA)	Q_a_ (mC)	Q_c_ (mC)
**PGE**	75.58 ± 15.44	82.45 ± 9.17	1.29	0.96
**IL-PGE**	96.39 ± 14.27	90.91 ± 7.41	1.33	1.02
**25 µg/mL CoPc-IL-PGE**	100.96 ± 10.56	88.43 ± 7.06	1.38	1.11
**50 µg/mL CoPc-IL-PGE**	104.60 ± 14.22	90.55 ± 11.17	1.50	1.24
**100 µg/mL CoPc-IL-PGE**	96.00 ± 15.70	82.59 ± 11.82	1.44	1.19
**200 µg/mL CoPc-IL-PGE**	96.72 ± 11.65	81.50 ± 10.18	1.50	1.22

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
