# Peer review of "Cobalt Phthalocyanine-Ionic Liquid Composite Modified Electrodes for the Voltammetric Detection of DNA Hybridization Related to Hepatitis B Virus"

_micromachines, 2021, doi:10.3390/mi12070753_

Round 1

Reviewer 1 Report

This work focuses on the electrochemical determination of the Hepatitis B virus based on cobalt phthalocyanine and ionic liquid modified pencil graphite electrodes. The scientific content of the paper poorly meets the academic standards and I highly recommend the re-organization of the manuscript. There are too many grammatical mistakes that make this paper difficult to understand. As in this version, I recommend rejection for numerous technical and scientific reasons. The manuscript needs a complete rewriting, properly proofreading and intensive double-checking of results, references and concepts before resubmission.

Author Response

Micromachines (MDPI)

micromachines-1227150

Cobalt phthalocyanine-ionic liquid composite modified electrodes for voltammetric detection of DNA hybridization related to Hepatitis B virüs

14 th June 2021

The list of our answers to the comments of reviewers

Thank you for valuable comments of reviewer-1, reviewer-2 and reviewer-3. We revised/corrected our manuscript according to the each comments pointed by reviewers. The references list was updated according to the comments of reviewers. The revised/corrected parts in the manuscript were shown as yellow highlighted.

Comments from the editors and reviewers:

Reviewer #1: This work focuses on the electrochemical determination of the Hepatitis B virus based on cobalt phthalocyanine and ionic liquid modified pencil graphite electrodes. The scientific content of the paper poorly meets the academic standards and I highly recommend the re-organization of the manuscript. There are too many grammatical mistakes that make this paper difficult to understand. As in this version, I recommend rejection for numerous technical and scientific reasons. The manuscript needs a complete rewriting, properly proofreading and intensive double-checking of results, references and concepts before resubmission.

Answer: First of all, thank to the Reviewer #1 for the valuable comments on the revision of our manuscript. According to your comment as well as the comments of reviewer-2 and reviewer-3, the manuscript is revised carefully, and also the grammatical mistakes are corrected. The references list was updated according to the comments of reviewers.

Reviewer 2 Report

What is the added value of the present work in comparison to the references (24 and 25)?

The authors are required to make a comparison with previous papers regarding guanine oxidation and HPV detection such as https://doi.org/10.1002/elan.201501113 and   https://doi.org/10.1002/elan.201700462

How the authors select 5% of ionic liquid?

Ligne 187, please double-check 2.21 ± 0.23 mA or µA ?

Please explain clearly the procedure of DNA immobilization including the mechanism of reaction in the case of DNA oligonucleotides linked from 5’ end with different groups (-PO4, -NH2).

It seems that the observed variation indicated in Figure S3, Figure S4, Figure S5, Figure S6, and Table S1 are within the experimental errors and can not be ascribed to the probe concentration, the hybridization time, and the HBV target concentration.

Figure 3. No difference in terms of peak-to-peak separation and peak magnitude for both IL-PGE and CoPc-IL-PGE. Authors are required to discuss this point.

What is the role of CoPc in this work?

Author Response

Micromachines (MDPI)

micromachines-1227150

Cobalt phthalocyanine-ionic liquid composite modified electrodes for voltammetric detection of DNA hybridization related to Hepatitis B virüs

14 th June 2021

The list of our answers to the comments of reviewers

Thank you for valuable comments of reviewer-1, reviewer-2 and reviewer-3. We revised/corrected our manuscript according to the each comments pointed by reviewers. The references list was updated according to the comments of reviewers. The revised/corrected parts in the manuscript were shown as yellow highlighted.

Comments from the editors and reviewers:

Reviewer #2:

  • What is the added value of the present work in comparison to the references (24 and 25)?

Answer: First of all, thank to the Reviewer #2 for the valuable comments.

In reference 34, a polymerase chain reaction (PCR)-free technique for the effective detection of genomic length hepatitis B virus (HBV) DNA is described by Chen et al. The honeycomb-like barrier layer of an anodic aluminum oxide (AAO) film was used as the substrate of the sensing electrode. A gold film was sputtered onto the AAO barrier layer surface as the electrode, followed by electrochemical deposition of gold nanoparticles (GNPs) on the hemisphere surface. The authors presented a very time consuming procedure while using numerous chemicals for surface modification of electrode (i.e approximately 6 hours)  that was applied then for impedimetric detection of HBV. In comparison to the study of Chen et al.[34], we followed a much easier procedure herein to develop single-use Cobalt phthalocyanine-ionic liquid composite modified electrode that was applied for a label-free voltammetric detection of HBV by measuring direct guanine oxidation signal. In addition, we performed modification of electrode surface in a shorter time than the one  of Chen et al.[34].

In the study of Mashhadizadeh et al. [35], the authors presented carbon paste electrode modified with magnetite and gold nanoparticles by following the time consuming procedure (i.e, 4 hours). After the immobilization of a thiol linked DNA probe onto the modified electrode’s surface during 6 hours, the impedimetric detection of target HBV DNA was then performed. In comparison to the study of Mashhadizadeh et al. [35], the significant properties of single-use electrodes (i.e, PGEs in our study) have provided some significant advantages to our assay; such as robustness, easy to use, being single-use and providing faster and more sensible detection protocol in comparison to carbon paste electrode any other expensive metal oxide film substrates, etc. In addition, we performed modification of electrode surface in a shorter time than the one  of Mashhadizadeh et al. [35].

The required discussion is given at conclusion as follows:

“The significant properties of PGEs have provided some significant advantages, such as robust, easy to use, being single-use and providing faster and more sensitive detection protocol in comparison with glassy carbon electrode [46], gold electrode [47,48] and carbon paste electrode [35], which were used for the detection of HBV. On contrast to the studies of Chen et al [34] and Mashhadizadeh et al. [35], the preparation and modification of electrode surface results herein in a shorter time (i.e less than one hour).”

  • The authors are required to make a comparison with previous papers regarding guanine oxidation and HBV detection such as https://doi.org/10.1002/elan.201501113 and https://doi.org/10.1002/elan.201700462

Answer:  Thank you for comments.

 Eksin et al. (https://doi.org/10.1002/elan.201501113) developed  CHIT and CNF composite  modified pencil graphite electrodes (CHIT-CNF-PGEs) for electrochemical detection of surface confined DNA hybridization related to HBV DNA. The preparation of  electrodes were completed in 75 min in the study of Eksin et al. (https://doi.org/10.1002/elan.201501113) however  it took  45 min in this present study.

In the study of Eksin et al. (https://doi.org/10.1002/elan.201501113), probe immobilization onto the surface of CHIT-CNF-PGE was resulted in 1 h. After then probe modified electrodes were dipped into the sample containing the complementary of probe (i.e HBV DNA target) during 1 h hybridization time. Then, voltammetric transduction was performed to detect HBV DNA hybridization. In comparison to our assay, the LOD (10.88 µg/mL) was found 57 times higher than the one reported in our study as 190 ng/mL. Erdem et al. (https://doi.org/10.1002/elan.201700462) developed hydroxyapatite nanoparticles (HaNPs) modified PGEs in the presence of 15 min modification time of HaNPs onto electrode’s surface. Then these electrodes were applied for electrochemical monitoring of sequence selective DNA hybridization related to HBV via with differential pulse voltammetry (DPV) technique. The solution phase hybridization procedure was performed during 15 min, and then the HaNP-PGEs were dipped into the samples for 1 hour for the immobilization of hybrids onto the electrode’s surface. Then, voltammetric transduction was performed to detect HBV DNA hybridization based on the changes at guanine signal. In comparison to our assay, the LOD (1.07 µg/mL) was found 5.5 times higher than the one reported in our study as 190 ng/mL.

In addition, the comparison of results presented in our study over to the earlier studies of Eksin et al. and Erdem et al. is also given in Table (below) for  the notice of reviewer.

Table . The comparison of our study developed by CoPc-IL- PGE over to the studies of Eksin et al. and Erdem et al. presenting electrochemical detection of HBV by using CHIT-CNF-PGE and HaNPs-PGE.

Reference

Electrode

Analyte

Method

Hybridization procedure

Experiments on its implementation to the real sample

DL

Eksin et al. 2016

CHIT-CNF-PGE

HBV

DPV

step by step

-

10.88 µg/mL

Erdem et al. 2018

HaNP-PGE

HBV

DPV

in solution phase

-

1.07 µg/mL

This work

CoPc-IL-PGE

HBV

DPV

in solution phase

0.19 µg/mL in buffer medium

2.48 µg/mL in FBS medium

* Eksin, E.; Erdem, A. Chitosan-carbon Nanofiber Modified Single-use Graphite Electrodes Developed for Electrochemical Detection of DNA Hybridization Related to Hepatitis B Virus. Electroanalysis 2016, 28, 2514-2521.

* Erdem, A.; Congur, G. Hydroxyapatite Nanoparticles Modified Graphite Electrodes for Electrochemical DNA Detection. Electroanalysis 2018, 30, 67 – 74.

The required discussion is given at conclusion as follows:

“In present study, the DL was calculated as 0.19 µg/mL in buffer medium (1.224 pmol in 40 µL sample), that was found to be lower in contrary to the ones reported in previous studies [33,43,48,51]. In addition, the DL in FBS medium was also found to be 2.48 µg/mL (16 pmol in 40 µL sample). To follow the procedure of solution phase hybridization in our study provides a fast, efficient and more sensitive hybridization resulting with detection limit (0.19 µg/mL) that is 57 times lower in comparison to the one (i.e 10.88 µg/mL) in the study followed the step by step hybridization procedure for label free HBV DNA detection performed by CHIT-CNF-PGEs [33].

In another study based on solution phase hybridization applied for a label-free HBV DNA detection by HaNPs-PGEs [51], 5.5 times higher DL value (i.e 1.07 µg/mL) was   reported in comparison to the one (i.e, 0.19 µg/mL) achieved in the present study.”

  • How the authors select 5% of ionic liquid?

Answer:  The ionic liquid % was used as 5 %  regarding to our earlier studies  (Erdem et al.,Colloids and Surfaces B: Biointerfaces 2014, 114, 261– 268; Eksin et al. Electroanalysis 2013, 25, 2321 – 2329) that we developed different nanocomposites modified electrodes by using ionic liquid in the presence of chitosan.

* Erdem, A.; Muti, M.; Mese, F.; Eksin, E. Chitosan–ionic liquid modified single-use sensor for electrochemicalmonitoring of sequence-selective DNA hybridization. Colloids and Surfaces B: Biointerfaces 2014, 114, 261– 268.

* Eksin, E.; Muti, M.; Erdem, A. Chitosan/Ionic Liquid Composite Electrode for Electrochemical Monitoring of the Surface-Confined Interaction Between Mitomycin C and DNA. Electroanalysis 2013, 25, 2321 – 2329.

  • Line 187, please double-check 2.21 ± 0.23 mA or µA ?

Answer: Thank you for your valuable correction. The values were corrected as follows:

“After immobilization of 2.5 μg/mL fsDNA onto the surface of each electrode; PGE, IL-PGE, CoPc-PGE and CoPc-IL-PGE, the guanine oxidation signal was measured by DPV technique (Figure S1). The average guanine signal was obtained respectively as 2.29 ± 0.18 µA (RSD %, 8.20 %, n=3), 2.39 ± 0.35 µA (RSD %, 14.55 %, n=3), 2.21 ± 0.23 µA (RSD %, 10.45  %, n=3) and 2.53 ± 0.28 µA (RSD %, 11.12 %, n=3).”

  • Please explain clearly the procedure of DNA immobilization including the mechanism of reaction in the case of DNA oligonucleotides linked from 5’ end with different groups (-PO4, -NH2).

Answer: In our study, the effect of labelling of DNA oligonucleotides with different chemical groups (-NH2, -PO4) was investigated upon to electrode’s response. According to the results, the highest guanine oxidation signal was obtained with –NH2 linked DNA immobilized CoPc-IL-PGE. According to the results, the highest guanine oxidation signal was obtained with –NH2 linked DNA immobilized CoPc-IL-PGE due to binding the amino linked DNA probe by the electrostatic interaction (π-π) to the surface of CoPc–IL-PGE completely by negatively charged phosphate groups of IL with positively charged amino group of DNA probe [ref. 43].

In addition, this explanation was included into the results and discussion section of our revised manuscript as follows:

“According to these results, the highest guanine oxidation signal was obtained by using –NH2 linked DNA probe immobilized CoPc-IL-PGE due to the fact that efficient binding of amino linked DNA probe to the surface of CoPc–IL-PGE via the electrostatic interaction (π-π) that is occured completely by negatively charged phosphate groups of IL with positively charged amino group of DNA probe [43].”

  • It seems that the observed variation indicated in Figure S3, Figure S4, Figure S5, Figure S6, and Table S1 are within the experimental errors and can not be ascribed to the probe concentration, the hybridization time, and the HBV target concentration.

Answer: According to your comment, the experimental results were checked. The mistake at labels in Figure S3 with a and d was corrected. The figure 3 with  corrected labels was shown as follows:

Figure S3. DPVs representing the guanine oxidation signals observed after hybridization between (a) 0.25, (b) 0.5, (c) 0.75, (d) 1 µg/mL amino linked DNA probe and 5 µg/mL HBV target.

Table S1. The average guanine oxidation signal (n=3) measured in the presence of hybridization of DNA probe with 5 µg/mL HBV target in various concentrations of DNA probe.

Probe Concentration (µg/mL)

Guanine oxidation signal (µA)

RSD %

0.25

6.00 ± 0.86

14.28

0.50

6.29 ± 0.13

2.13

0.75

5.35 ± 1.14

21.40

1

6.17 ± 1.01

16.42

  • Figure 3. No difference in terms of peak-to-peak separation and peak magnitude for both IL-PGE and CoPc-IL-PGE. Authors are required to discuss this point.

Answer: According to Figure 2, the average value of Ia measured by IL-PGE and CoPc-IL-PGE was obtained as 96.39 ± 14.27 and 104.60 ± 14.22, respectively. As a result, the increase at Ia in comparison to the one of PGE was found to be 27.53 % and 38.40 % in the presence of IL-PGE and CoPc-IL-PGE, respectively.

As we reported in the manuscript that the surface areas of PGE, IL-PGE, CoPc-PGE and CoPc-IL-PGEs were calculated and found to be 0.228 cm2, 0.291 cm2, 0.163 cm2 and 0.316 cm2, respectively. The increase % at the surface area of CoPc-IL-PGE in comparison with PGE was calculated and recorded as 39%. As a result, the highest surface area was obtained by CoPC-IL-PGE of all.

“The surface areas of PGE, IL-PGE, CoPc-PGE and CoPc-IL-PGEs were found to be 0.228 cm2, 0.291 cm2, 0.163 cm2 and 0.316 cm2, respectively. For instance, the increase % at the surface area of CoPc-IL-PGE in comparison with PGE was calculated and recorded as 39%. As a result, the highest surface area was obtained by CoPC-IL-PGE of all.”

  • What is the role of CoPc in this work?

Answer: Metallophthalocyanines; such as CoPc have been used as the electron transfer mediators, and also to increase the electro-catalytic activity in the redox processes for the development of electrochemical (bio)sensors [references  1 to 7, given below].

As we meantioned in the manuscript, Cobalt phthalocyanine (CoPc) are two dimensional (2D) organic macrocyclic molecular catalysts with cobalt atoms at the center, that is similar to porphyrin in its structure and demonstrates certain physicochemical, unique electronic and optical properties.  

Due to the advanced properties of Cobalt phthalocyanine-ionic liquid composites accelerating the electron transfer and increasing the surface area of electrode, the voltammetric detection of HBV DNA resulted in our study with a lower detection limit with a good selectivity even in the FBS medium.

The required explanation was included into the conclusion section of our revised manuscript as follows:

“Due to the advanced properties of Cobalt phthalocyanine-ionic liquid composites accelerating the electron transfer and increasing the surface area of electrode, the voltammetric detection of HBV DNA resulted in our study with a lower detection limit with a good selectivity even in the FBS medium.”

References used for comments of reviewer-1:

 [1] Stefanov, C.; van Staden, J.K.F.; Stefan-van Staden, R.I. Review—Enzymatic and Non-Enzymatic (bio)sensors Based on Phthalocyanines. A Minireview. ECS J. Solid State Sci. Technol. 2020 9 051012.

[2] Abbas, M.N.; Saeed, A.A.; Ali, M.B.; Errachid, A.; Zine, N.; Baraket, A.; Singh, B. Biosensor for the oxidative stress biomarker glutathione based on SAM of cobalt phthalocyanine on a thioctic acid modified gold electrode. Journal of Solid State Electrochemistry 2019, 23, 1129–1144.

[3] Hosseini, H.; Mahyari, M.; Bagheri, A.; Shaabani, A. A novel bioelectrochemical sensing platform based on covalently attachment of cobalt phthalocyanine to graphene oxide. Biosensors and Bioelectronics 2014, 52, 136–142.

[4] Stefan-van Staden, R. I.; Ilie-Mihai, R. M.; Gugoasa, L. A.; Bilasco, A.; Visan, C. A.; Streinu-Cercel, A. Molecular recognition of IL-8, IL-10, IL-12, and IL-15 in biological fluids using phthalocyanine-based stochastic sensors. Anal. Bioanal. 2018, 410, 7723.

[5] Özcan, L.; Şahin, Y.; Türk, H. Non-enzymatic glucose biosensor based on overoxidized polypyrrole nanofiber electrode modified with cobalt(II) phthalocyanine tetrasulfonate. Biosens. Bioelectron. 2008, 24, 512.

[6]  Fan, Z.; Fan, L.; Dong, C. Talanta 2018, 198, 16.

[7] Arrieta, A.; Rodriguez-Mendez, M.L.; De Saja, J.A. Langmuir–Blodgett film and carbon paste electrodes based on phthalocyanines as sensing units for taste. Sens. Actuat. B Chem. 2003, 95, 357-365.

Reviewer 3 Report

  • Suggestions to Author/s 
  1. Dear Dr.Arzum Erdem and Dr. Ece Yaralı, as a selected reviewer, I made a prompt check of your article: »Cobalt phthalocyanine-ionic liquid composite modified electrodes for the voltammetric detection of DNA hybridization related to Hepatitis B virus« and found it (x) Excellent, accept the submission (5). Therefore, I will make the strong suggestion to the Journal’s editor, to accept it for publication as soon as it is possible.  

  1. DearDr.Arzum Erdem and Dr. Ece Yaralı, you are kindly suggested to cite the  

article: Zupan, S., Flipic, B., Babic, M. »Piezoelectric Immunosensors«, Acta Pharmaceutica, 1992, 42(1): 361-366.   

 3. You are kindly asked to correct the minor mistakes in your text like this: 

Line No.:       Correct (Add/ Delete)... 

  1. Add: the 
  1.    Delete: present 

     19     Delete: was   Add: were 

     22     Delete: proposed 

     23     Add: the 

     38     Add: its 

     48     Add: are Delete: is available;  

              Add: the data of the 

55.  Delete: have been;  Add: are 

58.  Add: the 

68.   Delete:; 

74.   Delete: a 

79.   Add: the 

87.   Delete: was; Add: are 

90.   Add: the 

91.   Delete: ; 

96.   Delete: at; Add: in the 

111. Delete: as seen 

119. Delete: using; Add: the 

124. Delete: optimum; Add: optimal 

125. Delete: were; Add: are 

126. Add: the 

143. Delete: were; Add: are 

146. Add: the; Add: of the 

158. Add: the ; Add: of the 

159. Delete: while; Add: during the 

164. Add: the 

203. Delete: were; Add: are 

204. Delete: procedure; Add: procedures 

205. Delete: ; 

230. Delete: optimum; Add: optimal 

235. Delete: was; Add: is 

248. Delete: ; 

255. Delete: was; Add: is 

266. Delete: ; 

282. Delete: ; 

285    Delete: ; 

Author Response

Micromachines (MDPI)

micromachines-1227150

Cobalt phthalocyanine-ionic liquid composite modified electrodes for voltammetric detection of DNA hybridization related to Hepatitis B virüs

14 th June 2021

The list of our answers to the comments of reviewers

Thank you for valuable comments of reviewer-1, reviewer-2 and reviewer-3. We revised/corrected our manuscript according to the each comments pointed by reviewers. The references list was updated according to the comments of reviewers. The revised/corrected parts in the manuscript were shown as yellow highlighted.

Comments from the editors and reviewers:

Reviewer #3: Dear Dr.Arzum Erdem and Dr. Ece Yaralı, as a selected reviewer, I made a prompt check of your article: »Cobalt phthalocyanine-ionic liquid composite modified electrodes for the voltammetric detection of DNA hybridization related to Hepatitis B virus« and found it (x) Excellent, accept the submission (5). Therefore, I will make the strong suggestion to the Journal’s editor, to accept it for publication as soon as it is possible.  

  • Dear Dr.Arzum Erdem and Dr. Ece Yaralı, you are kindly suggested to cite the article: Zupan, S., Flipic, B., Babic, M. »Piezoelectric Immunosensors«, Acta Pharmaceutica, 1992, 42(1): 361-366.

Answer: First of all, thank to the Reviewer #3 for valuable comments.

We included the following reference Zupan, S., Flipic, B., Babic, M. Piezoelectric Immunosensors, Acta Pharmaceutica, 1992, 42(1): 361-366” as reference 2 into  the revised introduction of our revised manuscript.

  • You are kindly asked to correct the minor mistakes in your text like this: 

 Line No.:       Correct (Add/ Delete)

  • Add: the 

16         Delete: present 

19         Delete: was   Add: were 

     22          Delete: proposed 

     23          Add: the 

     38          Add: its 

     48          Add: are Delete: is available;  

                   Add: the data of the 

     55          Delete: have been;  Add: are 

58          Add: the 

68          Delete:; 

74          Delete: a 

79          Add: the 

87          Delete: was; Add: are 

90          Add: the 

91          Delete: ; 

96          Delete: at; Add: in the 

111        Delete: as seen 

119        Delete: using; Add: the 

124        Delete: optimum; Add: optimal 

125        Delete: were; Add: are 

126        Add: the 

143        Delete: were; Add: are 

146        Add: the; Add: of the 

158        Add: the ; Add: of the 

159       Delete: while; Add: during the 

164       Add: the 

203       Delete: were; Add: are 

204       Delete: procedure; Add: procedures 

205       Delete: ; 

230       Delete: optimum; Add: optimal 

235      Delete: was; Add: is 

248      Delete: ; 

255      Delete: was; Add: is 

266      Delete: ; 

282      Delete: ; 

285     Delete: ; 

Answer: Thank you for your valuable corrections. According to the your comments, each of the minor mistakes that you pointed, was corrected as follows:

Line 3, “…electrodes for the voltammetric..”

Line 16, “…in this study,…”

Line 19, “…the electrochemical behaviour of electrodes were studied…”

Line 22, “The implementation of our biosensor to serum samples…”

Line 23, …using the fetal bovine serum (FBS)…” 

Line 38, “…porphyrin in its structure…”    

Line 48, “More than 200 million people are infected with HBV in the world according to the data of the World Health Organization (WHO) [18]."

Line 55, “There are earlier electrochemical studies…”        

Line 58, “The hybridization between the probe and…”

Line 68, “…using techniques CV, EIS…”       

Line 74, “…reference electrode (BAS, Model RE-5B, W. Lafayette, USA),  platinum wire…” 

Line 79, “…supplied with the NOVA 1.11 software package.”       

Line 87, “…their stock solutions are given in details…”         

Line 90, “Each of the following steps was performed…”

Line 91, “…perform the experimental procedure properly”

Line 96, “…in more details in the Supplementary Materials.”

Line 106, “Results and Discussion”

Line 111, “…was observed in Figure 1e and f.”

Line 119, “…by the CV technique (Figure 2).”       

Line 124, “Thus, 50 µg/mL was selected as optimum optimal concentration…”       

Line 125 and 126, “…the optimization studies are listed in the Table 1.”     

Line 143, “…CoPc-IL-PGE are presented…”

Line 146, “The effective surface areas of the each of the electrodes were calculated…”       

Line 158, “As well, the electrochemical characterization of the each of the electrode…”       

Line 159, “…EIS technique (Figure 4) during the investigating the change…”

Line 164, “After immobilization of the probe…”

Line 187, “…2.21 ± 0.23 µA (RSD%, 10.45%, n=3) and…”

Line 203, “…the results are presented in Figure S2.”

Line 204, “…by following two different procedures (i) step by step, or (ii) solution phase…”

Line 205, “…the hybridization efficiency on the response i.e guanine oxidation signal.”

Line 230, “Hence, 5 min was chosen as optimal hybridization time in our study.”

Line 235, “The resulting line graph is presented in Figure S6.”     

Line 248, “…in different dilution ratio (1: 100) or (1:500) for FBS: PBS.”

Line 255, “…in target concentration varying from 5 µg/mL to 35 µg/mL is shown in Figure S8.”      

Line 266, “…in the presence of unwanted substituents e.g, NC, MM sequences…”     

Line 282, “…on the hybridization process e.g, hybridization time, the concentrations of…”

Line 285, “…at both medium in PBS (i.e. 0.19 µg/mL) and…”

Line 299, “…perform the high hybridization process.”

Round 2

Reviewer 2 Report

Although the authors tried to answer all my comments, their answers still not convincing me. Furthermore, this work is not original, the authors published several papers almost similar to the present work. However, At this level, I should accept the publication of this manuscript in its present form.